# Innovative Design of Targeted Nanoparticles: Polymer–Drug Conjugates for Enhanced Cancer Therapy

**DOI:** 10.3390/pharmaceutics15092216

**Published:** 2023-08-27

**Authors:** Varaporn Buraphacheep Junyaprasert, Parichart Thummarati

**Affiliations:** Department of Pharmacy, Faculty of Pharmacy, Mahidol University, 447 Sri-Ayutthaya, Rajathavee, Bangkok 10400, Thailand

**Keywords:** polymer–drug conjugates, targeted nanoparticles, cancer therapy, EPR effect, passive targeting, active targeting

## Abstract

Polymer–drug conjugates (PDCs) have shown great promise in enhancing the efficacy and safety of cancer therapy. These conjugates combine the advantageous properties of both polymers and drugs, leading to improved pharmacokinetics, controlled drug release, and targeted delivery to tumor tissues. This review provides a comprehensive overview of recent developments in PDCs for cancer therapy. First, various types of polymers used in these conjugates are discussed, including synthetic polymers, such as poly(*↋*-caprolactone) (PCL), D-α-tocopheryl polyethylene glycol (TPGS), and polyethylene glycol (PEG), as well as natural polymers such as hyaluronic acid (HA). The choice of polymer is crucial to achieving desired properties, such as stability, biocompatibility, and controlled drug release. Subsequently, the strategies for conjugating drugs to polymers are explored, including covalent bonding, which enables a stable linkage between the polymer and the drug, ensuring controlled release and minimizing premature drug release. The use of polymers can extend the circulation time of the drug, facilitating enhanced accumulation within tumor tissues through the enhanced permeability and retention (EPR) effect. This, in turn, results in improved drug efficacy and reduced systemic toxicity. Moreover, the importance of tumor-targeting ligands in PDCs is highlighted. Various ligands, such as antibodies, peptides, aptamers, folic acid, herceptin, and HA, can be incorporated into conjugates to selectively deliver the drug to tumor cells, reducing off-target effects and improving therapeutic outcomes. In conclusion, PDCs have emerged as a versatile and effective approach to cancer therapy. Their ability to combine the advantages of polymers and drugs offers enhanced drug delivery, controlled release, and targeted treatment, thereby improving the overall efficacy and safety of cancer therapies. Further research and development in this field has great potential to advance personalized cancer treatment options.

## 1. Introduction

Cancer treatment poses a significant challenge to medicinal sciences. Although chemotherapy and radiation therapy are primary therapeutic strategies, they often cause severe systemic side effects [1]. In addition, the low solubility of many chemotherapeutics causes aggregation, triggering an immune response and clearance from the body. This ultimately decreases the circulation time in the bloodstream and reduces its effectiveness in delivering the free drug to tumor sites [2,3]. Polymer–drug conjugates (PDCs) are drug delivery technologies that were first initiated by Horst Jatzkewitz in 1955 [4]. Several drug molecules are covalently bound to polymeric carriers through bioresponsive linkers to improve stability with the diversity, specificity, and functionality of biomolecules [5]. PDCs offer various advantages for cancer therapy. They can improve drug solubility and loading capacity [6,7], improve pharmacokinetic profiles by controlling and maintaining drug release [8,9], and increase drug half-life by decreasing immune system recognition. In addition, they increase drug accumulation specificity at the target site through passive and active transport [10,11,12,13]. The design and synthesizing of new PDCs that can interact effectively with biological systems is a challenge. Drugs must have free functional groups that can be conjugated directly to polymer backbones through chemical linkers (Figure 1); otherwise, PDCs are impossible to form. For example, curcumin (CUR) presents the functional group of R-OH and R-C=O-R, as seen in Figure 1. These functional groups can be linked to R-C=O-OH and R-HN_2_ of the polymer to form ester and hydrazone linkers, respectively. PDCs also enable the codelivery of drugs and/or bioactive molecules with different properties in one nanoparticle, making them multifunctional [14,15]. Due to these advantages over the free form of a drug, PDCs have been widely applied in medicinal treatments for various diseases such as cancer, osteoporosis, infection, and immunodeficiency. The focus of this article is to review the rational design of PDCs for cancer therapy. PDCs of various chemistries and architectures have been discussed, with particular emphasis on ideas for enhancing PDC systems.

## 2. Research Progress on PDCs

The origin of PDCs began in 1955 when Horst Jatzkewitz developed polyvinylpyrrolidone (PVP) conjugated with the primary amine of glycyl-L-leucyl-mescaline using a dipeptide linker to enhance the antianxiety drug [4]. In 1958, Mathé et al. presented the important phenomenon of PDCs by pioneering the conjugation of drugs to immunoglobulin, thus setting the stage for traditionally targeted drug delivery systems [16]. Subsequently, a variety of PDC systems based on PVP of various antibiotic agents have been developed to provide sustained release of drugs into the bloodstream, selective targeting, and extended half-life, such as penicillin [17]. In 1974, De Duve et al. discovered that PDCs can be degraded by many enzymes localized in the lysosomal compartment of cells and the lysosomotropism of macromolecules [18]. Based on the interpretation of these results, Helmut Ringsdorf presented the foundation of PDCs for targeted drug carriers in 1975 [19]. Since then, PDCs have become a rapidly growing field, with nearly a dozen polymeric conjugates progressing to clinical trial studies. The first generation of PDCs has also found applications in disease treatment, such as the use of PEGylated protein of bovine-serum-albumin-conjugated polyethylene glycol (PEG) [20,21]. Subsequently, PDCs gained considerable attention in cancer therapy between 2000 and 2010, with the aim of selective accumulation in tumor tissues. This approach was fascinating for targeted drug delivery, reducing systemic side effects. During this period, poly(*N*-hydroxypropyl methacrylamide) (HPMA) copolymer-based drug conjugates were evaluated in clinical trials. Notably, HPMA-doxorubicin (DOX) conjugates exhibited enhanced anticancer activity and decreased side effects, progressing to clinical trial stages [22,23,24]. The first PDC used for cancer therapy was PEGylated liposomal DOX (Doxil^®^), which was approved by the FDA in 1999 and by the European Medicines Agency (EMA) in 2000 as a single agent for the treatment of patients with advanced ovarian cancer who did not receive platinum-based first-line treatment [25]. Following the 2010s, targeted therapies made significant advances in revolutionizing cancer treatment. Some noteworthy therapies are the use of polyglutamic acid (PGA) conjugated with paclitaxel (PTX), camptothecin (CPT) (CT-2106) conjugate, and a polystyrene-maleic anhydride-neocarzinostatin conjugate. These conjugates have been approved for the treatment of hepatocellular carcinoma in Japan [26]. These approaches were developed to identify specific molecular targets in tumor tissues, resulting in improved outcomes and reduced side effects compared to traditional therapies. In addition, the combination of different drugs within polymer conjugates was studied to achieve synergistic effects or target multiple pathways involved in cancer growth and progression. This innovative approach enabled more effective treatment strategies and holds the potential to reduce drug resistance. The history of the progression of the research on PDCs is shown in Figure 2.

## 3. The Principle of PDCs

The principle of PDCs was first revealed by Helmut Ringsdorf, as previously mentioned. In this model, it was envisioned that the drug-attached polymeric carrier could not only be modulated, but also that active targeting could be achieved by introducing a homing moiety into the same polymeric carrier [19]. The compositions of the PDCs are divided into three different units: i.e., a solubilizing zone, drug–polymer chemical linkers, and a transport system (Figure 3). The first unit of a polymer is used to solubilize all of the macromolecules without toxicity. The second unit is the drug linked to the polymer via a chemical linker. The last unit is the area of the targeting ligand located in the hydrophilic region of the polymer. It can enhance the ability to carry the entire macromolecule to biological target sites. Many publications have attempted to develop these three parts, with the expectation that PDCs may provide some benefits, such as enhanced drug solubility and activity, modified pharmacokinetic profile, reduced toxicity, polymer-specific effects, and drug combination along the polymer chain. The task of acquiring successful PDCs seems complex because various factors affect polymers and nanoparticles. The selection of polymeric macromolecular carriers, the desired target (intracellular, lymphatic system, etc.), the type of conjugation (direct or indirect), the chemistry of the linker, and the molecular weight (MW) are considered key parameters [27]. Moreover, it should be noted that the design of an appropriate polymeric carrier must be strongly influenced by its proposed route of administration and frequency of dosing.

## 4. PDC Development for Cancer Treatment

### 4.1. Modified Physicochemical Properties of Polymers

For decades, PDCs have attracted considerable attention as a means of delivering drugs or bioactive molecules. Polymers play a dominant role in making the whole macromolecule of PDCs soluble. Generally, PDC-based polymeric nanoparticles are composed of a hydrophobic core and a hydrophilic shell that can self-assemble to form nanoparticles [28]. The inner core can be used as a storage site for hydrophobic molecules, which helps to increase the aqueous solubility of hydrophobic drugs. Meanwhile, the outer shell can help to improve stability by protecting active molecules (both hydrophobic and hydrophilic molecules) from interactions with blood components, reducing recognition by the reticuloendothelial system (RES) and enzyme degradation, and delivering the drug to intracellular sites of action [27,29]. This approach is known as “stealth nanocarriers”.

Many degradable polymers for PDC systems have been studied, both natural and synthetic polymers. In short, the polymers used for PDCs should be biodegradable, biocompatible, nontoxic to the human body, and completely eliminated from the body. Moreover, they should possess functional groups that can be covalently bound with biologically active molecules through a bioresponsive linker, as exemplified in Figure 4. Several types of polymers have been employed for PDCs such as PEG, PVP, HPMA, poly(*↋*-caprolactone) (PCL), poly(lactic-co-glycolic acid) (PLGA), D-α-tocopheryl polyethylene glycol (TPGS), hyaluronic acid (HA), dextran, alginate, pectin, and starch [27,30]. In the first generation of PDCs, PEG-L-asparaginase (ASP) (Oncaspar^®^) and HPMA-DOX (PK1, FCE28068) were evaluated in clinical studies [22,28,30,31]. This peptidyl linker was designed to hydrolyze via thiol-dependent proteases after lysosomotropic delivery. The second-generation PDCs based on combinations of high-molecular-weight HPMA copolymers, a glycyl-phenylalanyl-leucyl-glycine (Gly-Phe-Leu-Gly) linker, and TNP-470 have been developed to enhance the selectivity of anticancer agents in tumor vessels, showing considerable promise in vivo [31].

The chemical linker of the drug plays an important role in its conjugation with the polymers, which can change the therapeutic potency of linked drugs or active molecules. The developed PDCs should preferably have a short and simple chemical structure. Bioactive agents are often conjugated to biocompatible polymer backbones via strongly biodegradable linkages, such as ester bonds, which are easily hydrolyzed in the presence of the esterase enzyme. However, some acid-labile linkers have been used to synthesize various pH-responsive PDCs for cancer therapy, such as disulfide [32], acetal [33], hydrazone [34], orthoester [35], and amide bonds [36] (Figure 4). These linkers provide stability in blood circulation, a higher drug load, and controlled and sustained release without a burst effect [27]. The obtained prodrugs have the ability to selectively release the active drug within the acidic conditions of tumor tissues or intracellular endosomes, occurring in a pH range of 4.5 to 6.5. Li et al. [37] developed acetal-linked polymeric micelles for enhanced CUR delivery. The in vitro result showed that the acetal-linked micelles exhibited a pH-dependent drug release behavior, which released faster at acidic pH (pH 5.0 and 6.0) but showed retardation of release at physiological pH. In our study, it was found that the PDCs of Gemcitabine (GEM)- and CUR-conjugated HA using the hydrazone linker were specific and fast in the acidic microenvironment (pH 5.0–6.5) while retarded in physiological pH (pH 7.4) [34], resulting in enhanced antitumor efficacy and improved drug safety.

Some polymers were unable to attain the desired properties because they face limitations such as high hydrophobicity, high crystallinity, and lack of active sites for drug conjugation. Furthermore, PDCs containing a single bioactive agent often have limitations in terms of clinical application prospects due to inadequate anticancer efficacies and acquired drug resistance of cancer cells. Many studies have tried to overcome these problems by modifying and combining two or more polymers, as well as decorating small molecules or targeting ligands. In this review, we selected four examples of polymers that have been widely studied in the delivery of anticancer drugs, and discuss strategies for the improvement of polymer properties from previous reports, including the findings from our research groups.

#### 4.1.1. PCL

PCL is a saturated aliphatic polyester polymer that was studied as early as the 1930s [38]. It can be synthesized via ring-opening polymerization of *↋*-caprolactone (CL) using a variety of anionic, cationic, and coordination catalysts such as stannous octoate, or via free-radical ring-opening polymerization of 2-methylene-1-3-dioxepane (MDO) (Figure 5). Each method affects the resulting MW, MW distribution, end-group composition, and chemical structure of the copolymers [39]. PCL offers several advantages in PDCs, such as a slower degradation rate in vivo and is biodegradable and biocompatible with the human body, with a melting temperature of 60 °C and a decomposition temperature of 350 °C [40,41]. Because the linkage along PCL is an ester bond that can be hydrolyzed under physiological conditions and eliminated from the body, these properties make PCL an attractive option for an efficient drug delivery system. However, it still has some limitations for PDCs and medical use due to its high hydrophobicity and crystallinity, leading to slow elimination from the body [42]. Furthermore, the degradation products resulting from the breakdown of PCL can potentially cause inflammation or adverse reactions in some individuals, leading to limitations in FDA approval. The degradation rate of PCL related to adverse reactions depends on its molecular weight, crystallinity, and other factors [43]. Therefore, thorough biocompatibility studies are essential to assess any potential risks associated with the use of PCL-based materials in clinical trials. Many reports have been designed by combining PCL with other hydrophilic molecules or polymers to modulate the physical and mechanical properties of PCL. For example, PCL-mPEG, PCL-D-α-tocopheryl polyethylene glycol 1000 succinate (TPGS), PCL-polyethylene oxide (PEO), PCL-polylactic acid (PLA), and PCL-PVP enhance the attractive properties of PCL, such as the improved elasticity, higher hydrophilicity, stealth properties, and faster degradation times, allowing much wider applications of the polymer in pharmaceutical and medical fields [44,45,46,47,48]. Furthermore, diblock copolymers and triblock copolymers such as PCL-PEG and (P(CL)_2_)-PEG, respectively, have been synthesized, showing different assembly behaviors, drug loading properties, and cellular uptake behavior [49,50]. Issarachot et al. reported that the diblock copolymer of PCL-PEG was more flexible and showed less crystallization compared to the triblock copolymer of (P(CL)_2_)-PEG [50]. The design of PCL-based copolymers for medical applications is summarized in Table 1. Despite the notable advances in PCL-based drug delivery systems, there are still challenges to be addressed [51].

#### 4.1.2. TPGS

TPGS is a water-soluble derivative of natural vitamin E, which is formed via esterification with PEG (MW of 1000). The structure is shown in Figure 6. It is composed of hydrophobic and hydrophilic segments in its structure, which present amphiphilic properties. Therefore, it has been widely used in pharmaceutically safe adjuvants as a wetting agent, emulsifier, stabilizer, and solubilizing agent [54]. Recently, TPGS has become more attractive in the field of drug delivery systems as a nanocarrier because it can improve the solubility and bioavailability of poorly water-soluble and poorly absorbed drugs [55,56,57,58]. Its safety has been reported, with the oral 50% lethal dose (LD_50_) being >7 g/kg for young adult rats of both sexes [59]. In addition, the US FDA has approved TPGS as a safe and biocompatible adjuvant. Most reports showed that TPGS has been prepared in prodrugs, in which TPGS is conjugated with drugs to improve the pharmacokinetic profile of drug molecules. Mi et al. reported the synthesis of the TPGS-cisplatin (CIS) conjugate, which exhibited pH-dependent drug release, much higher cellular uptake, and higher cellular cytotoxicity compared to the unconjugated drug [60]. Our previous work also developed targeted PDCs using folic-acid-conjugated TPGS to deliver methotrexate (MTX). The results showed that these copolymers potentiated cytotoxicity and cellular uptake efficiency for breast cancer cells [61]. Some publications conjugated TPGS with other polymers to improve their properties. For example, TPGS-*b*-PCL copolymers have been utilized for drug delivery, with the aim of achieving the combined benefits of TPGS and PCL to increase the hydrophobicity of the copolymer and help with water-insoluble drugs. These copolymers have shown successful applications in cancer therapy by increasing drug loading and cytotoxic activity in liver cancer [62]. Another example is chitosan conjugated with TPGS and further decorated with transferrin, which was used to form targeted nanocarriers to deliver docetaxel (DTX). It provided a bioadhesive property and cytotoxicity that were useful for brain cancer therapy [63]. Nowadays, TPGS is widely investigated to overcome multidrug resistance (MDR) because TPGS has shown inhibitory activity to P-glycoprotein (P-gp) and potent antitumor activity, resulting in enhanced bioavailability of drugs such as MTX, DTX, DOX, CIS, and PTX [64,65,66]. Almawash et al. [67] successfully boosted the cytotoxicity of MTX by using PLGA-TPGS. The results revealed that the conjugation of MTX-PLGA-TPGS provided an improved pharmacokinetic profile and increased drug stability in the blood circulation as a result of the properties of TPGS. This led to increased cellular uptake and improved drug efficiency for cancer treatment, such as antibodies [68], galactosamine [69], and folic acid [61]. In addition, some targeting ligands can be decorated on TPGS to enhance cellular uptake. Gan et al. [68] investigated novel sorafenib (Sf)-loaded polymeric nanoparticles for the targeted therapy of hepatocellular carcinoma. Anti-GPC3 antibody (Ab) and Sf were grafted onto a TPGS-PLC block copolymer, which was further self-assembled from nanoparticles. The result showed that NP-Sf-Ab showed robust stability and achieved excellent Sf release in the cell medium. The MTT assay confirmed that NP-Sf-Ab caused much higher cytotoxicity than non-targeted NP-Sf and free Sf. Finally, NP-Sf-Ab was shown to greatly inhibit tumor growth in HepG2-xenograft-bearing nude mice without obvious side effects. Examples of useful TPGS based on PDCs for cancer drug delivery are listed in Table 2.

#### 4.1.3. PEG

PEG, a hydrophilic polymer, is a common component widely used in the development of PDCs and nanoparticles for cancer treatment [75]. PEG is made up of an ethylene glycol (EG) subunit which is surrounded by two to three water molecules, providing information about the shell around the micelles. The structure is shown in Figure 7. PEGylation, which involves the covalent binding of anticancer drugs or bioactive molecules with PEG polymers, offers promising carriers for cancer therapy. These carriers can improve the pharmacokinetics and biocompatibility of drugs, as well as enhance their circulation time in the body. PEGylation offers several benefits, including reduced clearance by the reticuloendothelial system (RES), increased tumor accumulation, and decreased toxicity to healthy tissues. In the case of PDCs, PEGylation of the polymer backbone or the drug molecule itself can lead to improved stability, solubility, and specificity to cancer cells. For example, PEGylated liposomal DOX (Doxil^®^) is the first PDC to show significant efficacy in the treatment of various cancers [25]. PEGylated nanoparticles have been extensively studied for cancer treatment, including drug delivery, imaging, and photodynamic therapy. The hydration sheath of PEG shells creates a steric barrier that prevents biomacromolecules from penetrating the polymer layer. PEG chains bind to the core through hydrophobic or electrostatic interactions, resulting in improved stability of the active molecules. Thus, numerous researchers have utilized these properties to develop stealth drug nanocarriers with the aim of prolonging the circulation time and reducing recognition and clearance via the mononuclear phagocyte system (MPS) in biotechnology therapeutics [76]. For example, novel dual-sensitive polypeptide-based CPT micelles conjugated with PEG showed sustainable drug release under physiological conditions and were able to enhance cellular internalization in human large lung cancer cells [77]. In a separate study, the conjugation of luteinizing hormone-releasing hormone (LHRH)-conjugated PEG-coated magnetite nanoparticles enhanced the hydrophilicity and biocompatibility of the nanoparticles. The conjugation led to the formation of a hydration sheath, resulting in improved stability in cell culture medium with minimal aggregation [78]. There are certain limitations associated with PEG. PEGylation can restrict the therapeutic efficacy of the conjugated product because its functionalization is limited to PEG chain ends, resulting in low drug loading capacities, a lack of amphiphilic properties, and nonspecificity. Some publications reported on the modification of amphiphilic polymers via PEGylation. For example, PEG-PLGA nanoparticles loaded with PTX had shown enhanced drug loading and release and improved tumor growth inhibition in animal models of breast cancer and lung cancer. In animal models of prostate cancer and lung cancer, the incorporation of miR-532-3p into vitamin B12-conjugated PLGA-PEG nanoparticles has shown improved efficacy in chemotherapy [79]. Furthermore, herceptin-conjugated PTX-loaded PCL-PEG worm-like nanocrystal micelles have shown enhanced drug loading and greater efficacy of chemotherapy in animal models of prostate cancer and lung cancer [80]. In summary, PEG can be used in a controlled-release system or as a PDC to improve the pharmacokinetic properties and efficacy of drugs used in cancer treatment.

#### 4.1.4. HA

Over the past decade, naturally occurring polymers have overwhelmingly been used for the development of polymeric nanoparticles. They show much better biodegradability in the biological system than synthetic polymers, thus preventing the accumulation of polymers in the body or within cells [81]. Among naturally occurring polymers, HA, a naturally highly hydrophilic mucopolysaccharide polymer, has gained much attention in drug delivery due to its biodegradability, biocompatibility, low toxicity, high potential for drug loading, and ease of chemical modification [82]. HA is composed of repeating disaccharide units of D-glucuronic acid (GlcUA) and N-acetyl-D-glucosamine (NAG), which are linear polyanions linked via β-1,3- and β-1,4-glycosidic bonds, respectively (Figure 8) [83]. The number of repeated disaccharides in a completed HA molecule can reach 10,000 or more. Its molecular mass is ~4 MDa, and each disaccharide is ~400 Da [83,84]. The average length of a disaccharide (one repeating unit) is about 1 nm. In solution, the HA chains entangle with each other at low concentrations in the form of an expanded random coil. In high-concentration solutions, HA forms molecular networks with shear-dependent viscosity, known as a hydrogel, in which drugs can be loaded either via physical entrapment or via covalent linkage. When pressure is applied, it easily moves and can be administered through a small-bore needle. Therefore, it is called a pseudoplastic material, which is an ideal lubricant [83].

HA is present in high concentrations in numerous malignant tumors compared to normal tissues, which is associated with tissue inflammation, angiogenesis, tumor invasion, and P-gp-mediated MDR [85,86,87]. It generally interacts with cells in at least two ways. First, it can bind to receptors on the surface of cancer cells, such as cluster of differentiation protein 44 (CD44), the receptor for HA-mediated mobility (RHAMM), lymphatic vessel endothelial receptor 1 (LYVE-1), and IVd4 and LEC receptors. Second, it has the ability to provide sustained attachment to hyaluronan synthase across the plasma membrane [81]. Among these, CD44 is the most studied HA receptor. CD44 is found to be overexpressed in various tumors, such as lung [88], breast [89], colon [90], stomach [91], and pancreatic cancers [92,93], while it is expressed in low levels in normal tissues. Due to the fact that CD44 and RHAMM are HA-binding receptors that are highly present on the surface of cancer cells, it is possible that HA may provide an ideal targeting ligand for selective binding to malignant tissues. Although HA has not been approved by the FDA for cancer therapy, it is widely used in PDC research for anticancer drug delivery systems due to its biocompatibility, biodegradability, nontoxicity, nonimmunogenicity and numerous modification sites [94]. Taking into account chemical structures, HA can be directly associated with drugs or through drug carriers via various linkers due to the high presence of carboxyl and hydroxyl groups. When hydrophobic molecules are conjugated with HA, they tend to easily form micelles or nanoaggregates with the drug inside and a hydrophilic HA shell layer [6,7,95]. Therefore, the formation of HA-drug conjugates provides several advantages: (i) they act as a hydrophilic carrier for the delivery of insoluble drugs, which enhances solubility and bioavailability; (ii) they help to protect drugs from deactivation and preserve their activities during circulation, leading to an improvement in the half-life of the drug in blood plasma and slowing clearance out from the body; and (iii) they actively target the drug specifically at the site of action, as mentioned earlier [96]. In general, HA has the benefit of being used as an active target carrier for active compounds, including anticancer drugs. Many previous studies reported the use of HA for conjugation with various anticancer drugs such as PTX, DOX, CPT, CIS, and QCT. The results revealed that HA–anticancer drug conjugates provide an increase in the solubility, stability, efficacy, and specificity of anticancer drugs [97,98,99]. Examples of HA–anticancer drug conjugates are summarized in Table 3.

### 4.2. Increased Drug Solubility and Loading Capacity

Poorly water-soluble drugs encounter challenges in pharmaceutical delivery, which impact the therapeutically effective concentration at the target site [104]. Nanotechnology has been used to overcome this problem because it can enhance drug solubility and loading capacity. Generally, drugs can be entrapped into nanocarriers via physical entrapment and chemical conjugation. Physical entrapment means that hydrophobic drugs are incorporated into the hydrophobic core via intermolecular forces such as hydrogen bonds, π-π interactions, and dipole–dipole interactions [105]. Although physical entrapment is simple and convenient, it is insufficient for drug loading capacity and faces the problem of burst release [106]. In the case of PDCs, bioactive agents are often linked to biocompatible polymer backbones using strongly biodegradable linkages, such as ester, amide, hydrazone, and acetal linkers. This strategy results in enhanced drug solubility, increased drug loading capacity, and controlled, sustained drug release without a burst effect [50,107,108]. It is important to select highly hydrophilic polymeric carriers to improve water solubility, such as PEG and polysaccharides. In the field of anticancer drug delivery, PDCs provide many benefits: not only increased solubility, but also decreased side effects. The solubility of PEGylated PTX, a conjugate of PTX and PEG, could be enhanced to about 1800 times that of the aqueous PTX solution (3665 μg/mL and 2 μg/mL, respectively) [11]. The conjugation of PEG and adenosine deaminase could enhance drug loading capacity and efficacy both in vitro and in vivo [109,110]. Henne et al. [111] developed PEGylated CPT by using a disulfide linker, which is an enzymatically cleavable linker. Folic acid was used as a targeting ligand in this study. The result revealed that it could self-assemble to form nanoparticles. Moreover, it showed better solubility than the aqueous solution, higher stability, and the best selective cleavage during circulation. A hydrophobic polymer core is separated and stabilized through a hydrophilic corona. In our previous study, we investigated the effect of the chemical linker and mol% CUR on the physicochemical properties of CUR nanoparticles for cancer therapy. CUR was conjugated on the HA backbone using ester and hydrazone linkers to form C*e*H and C*h*H, respectively [112]. The result revealed that HA increased the solubility of CUR in both nanoparticles. An increase in mol% CUR on the HA backbone resulted in the failure to form nanoparticles because the hydrophobic property was too high to produce nanoparticles. Subsequently, these C*h*H nanoparticles were further conjugated with GEM for codelivery to cancer cells. The result showed that GEM did not interrupt the solubility of CUR but improved the nanoparticle characteristics of the nanoparticles by decreasing the size and distribution [34]. In other studies, copolymers have been widely developed for improved amphiphilic properties, as they directly affect the in vivo circulation and biological activity of PDCs [113]. Tang et al. [114] developed DOX-HPMA conjugates to form self-assembled nanoparticles, and compared a linear one with a core cross-linked one. In vitro studies revealed that both showed slower drug release and improved solubility and stability. In contrast, in vivo pharmacokinetic behavior studies showed that the cross-linked copolymer nanoparticles resulted in good blood stability and long-lasting circulation time compared to those of the linear block copolymer nanoparticles and the free drug. Examples of PDCs that improve drug solubility are shown in Table 4.

### 4.3. Modified Drug Release and Controlled Delivery

Numerous available research articles report the potential of PDCs to provide unique polymer properties for the controlled and sustained release of bioactive agents [116]. Modification of the controlled and sustained release profile provides several benefits, including enhancing the drug accumulation at target sites, preventing the burst release of the drug in the bloodstream, and facilitating targeted drug delivery. This leads to increased efficacy and decreased toxicity of the drug or bioactive molecules. In short, the drug release rate depends on the covalent bond between the drugs and the polymer/nanocarriers [117]. The bonding between the drug and polymers should be stable in the blood circulation to protect the drug and prevent burst drug release. However, they should be hydrolyzed to release the drug based on physiological needs, following the normal physiological process of the stage of the disease. Systems have the ability to undergo dramatic chemical or physical changes in physiological responses to internal stimuli such as pH, redox, ionic strength, temperature, and lysosome/enzymes and external stimuli that can induce a response via stimuli-generating devices, such as pulsed drug delivery, such as electric, magnetic, and ultrasonic [118,119,120]. In this review, we focus on internal-stimuli-responsive drug delivery.

#### 4.3.1. pH-Responsive Drug Delivery

Many pH-responsive PDCs have been developed for anticancer and noncancer applications because they can be used as a trigger for drug release related to physiological conditions. The pH-sensitive bond is used to covalently link between drugs and polymeric carriers [117]. These systems have been intensively investigated for the delivery of anticancer drugs due to their enormous improvement in specificity and efficacy. It is known that the pH of normal and cancer tissues is different. Compared to normal tissues, the pH of the tumor tissue environment and intracellularly in the lysosomes is slightly acidic (pH_normal cells_ = 7.20–7.45, pH _cancer environments_ = 6.50–6.90, and pH_lysosomes_ = 4.5–6.5) [121,122]. Therefore, pH-sensitive linkers should be cleaved under acidic conditions, such as the pH of tumor tissues and lysosomes, while they are stable at physiological pH. Many acid-labile linkers have been reported to improve drug release profiles, such as hydrazone, amide, imine, cis-acotinyl, oxime, ketal, and acetal, which are shown in Figure 9. The favorite pH-responsive chemical bond, which has been extensively explored, is the hydrazone bond, because of its acute responsiveness in drug delivery behavior. Our study showed that the CUR-hydrazone bond (C*h*H) had better nanoparticle characteristics, including critical aggregation concentration (CAC), particle size, and stability, at physiological pH than nanoparticles with an ester bond (C*e*H), because C*h*H had a more flexible and less-bulky structure than C*e*H. Furthermore, C*h*H could better retard the release of CUR than C*e*H in physiological pH, but there was faster release under tumor conditions [112]. Jiang et al. [123] developed amphiphilic polycarbonate conjugates of DOX with hydrazone linkers. The release of DOX in an acidic environment (pH 5.0) was faster than that at neutral pH (pH 7.4). Our group also developed polymeric nanoparticles of HA-conjugated CUR and GEM using hydrazone bonds for cancer therapy [34]. The GEM and CUR release profiles of the nanoparticles were specific and fast in the acidic microenvironment (pH 5.0 to 6.5) while retarded at physiological pH (pH 7.4), indicating a dependence on pH. An in vitro cytotoxicity study showed that GEM-HA-CUR nanoparticles had higher toxicity and synergistic effects in PANC-1, A549, Caco-2, and HCT116 cells. Other chemical bonds can also be used to prepare pH-responsive PDCs. Li et al. [37] developed acetal-linked PDCs using mPEG-PLA to deliver CUR, which showed faster release at lower pH values (pH 5.0–6.0). The prepared PDCs could enhance the cytotoxicity of human hepatocellular liver carcinoma. The process of PDCs delivered to the target site is shown in Figure 10.

#### 4.3.2. Enzyme-Responsive Drug Delivery

The high expression of specific enzymes in certain diseases provides an advantage in using PDCs as triggers for drug release at the target site. Drugs are chemically linked to polymer carriers through enzyme-responsive bonding that can be cleaved by specific enzymes at the target site [124]. Recently, significant attention has been focused on designing and developing enzyme-responsive drug delivery systems, especially anticancer drug delivery systems. Legumain is highly overexpressed in most solid human tumors, making it a potential trigger. Shi et al. [125] developed an octapeptide, glycine-cysteine-glycine-alanine-alanine-asparagine-leucine-glutamic acid (Gly-Cys-Gly-Ala-Ala-Asn-Leu-Glu), attached to PEG-based PDCs, which was specific for legumain. The synthesized polymer-CIS conjugates with such peptide linkers exhibited great potential for increased stability in plasma and enhanced gastric cancer therapy. Zhang et al. [126] developed PEGylated GEM dendrimers with glycyl-phenylalanyl-leucyl-glycine (Gly-Phe-Leu-Gly) tetrapeptide as a spacer. This spacer can be cleaved by cathepsin B, which is much lower in normal cells than in tumor cells [127]. The results suggested that more than 80% of GEM was released from dendrimers under lysosomal cysteine protease cathepsin B conditions but showed lower release in the absence of cathepsin B, indicating that dendrimer-GEM could maintain stability in blood circulation. In vitro and in vivo studies confirmed its improved antitumor efficacy and reduced side effects in normal tissue compared to the GEM solution. Another study on non-anticancer drugs reported by Shivhare et al. indicated that ornidazole-conjugated inulin-based peptides could be cleaved by specific inulinases [128]. The resulting core–shell nanostructures could encapsulate ornidazole in the hydrophobic core and rapidly release it in the presence of an inulinase enzyme. Other triggers, matrix metalloproteinase 2 (MMP-2) and MMP-9, are widely known as tumor-associated enzymes. They are involved in many physiological and pathological processes, including tissue development, wound healing, and cancer progression. Chau et al. developed dextran−peptide−MTX conjugates using proline-valine-glycine-leucine-isoleucine-glycine (Pro-Val-Gly-Leu-Ile-Gly) as the peptide linker for tumor targeting via MMP-2 and MMP-9. The peptide linker was stable in the systemic circulation, but cleaved to release peptidyl MTX due to the presence of MMP-2 and MMP-9 [129]. Examples and therapeutic applications of smart enzyme-responsive polymers and their applications are presented in Table 5.

#### 4.3.3. Temperature-Responsive Drug Delivery

Temperature-responsive polymer drug conjugates have been extensively studied for cancer drug delivery. These polymers undergo a reversible phase transition from a hydrophilic to hydrophobic state in response to an external temperature stimulus, allowing the controlled release of drugs at elevated temperatures associated with cancer tissues as a trigger to cleave the linker between the polymer and the drug [134]. Because of its temperature sensitivity, poly(*N*-isopropylacrylamide) (PNIPAM) is a commonly used temperature-responsive polymer for drug conjugation in cancer therapy. The lower critical solution temperature (LCST) of PNIPAM is approximately 32 °C. One study developed PNIPAM-based copolymer conjugates with DOX for targeted delivery to breast cancer cells. The copolymers were designed to have LCSTs lower than the physiological temperature, allowing drug release within cancer cells upon endocytosis. In vitro studies showed that PNIPAM-DOX conjugates were more effective in killing cancer cells than free DOX, while showing reduced toxicity to normal cells [135]. Another study reported the development of a poly(vinyl caprolactam) (PVCL)-based drug delivery system for DOX, a chemotherapy drug commonly used in the treatment of cancer. The researchers synthesized a PVCL-DOX conjugate by covalently linking doxorubicin to PVCL through a pH-sensitive linker. The resulting conjugate showed a temperature-triggered release of doxorubicin at the tumor site in vitro and in vivo, indicating its potential as a targeted drug delivery system for cancer treatment [136,137]. Other polymers that have been studied for this application include poly(*N*,*N*-diethylacrylamide) (PDEAA), PVP, and poly(*N*-vinylacetamide) (PNVA).

### 4.4. Improved Drug Stability under Physiological Conditions

Improvement in drug stability under physiological conditions using PDCs is one approach to overcome the problem of drug instability. CPT is a potent antitumor agent for colon and gastric cancer, but has limitations due to its low aqueous solubility and poor stability both in vitro and in vivo [138]. The active form of CPT, CPT-lactone, is present at pH 4.5, but is unstable under physiological conditions, where its lactone ring readily opens and converts to CPT-carboxylate (Figure 11) [138]. Various PDCs for CPT have been developed via conjugation with polymer backbones to increase solubility and stability through the OH group of the lactone ring. Several polymers, including PEG [139], PEG-methyl ether methacrylate (MA) [140], cyclodextrin [141], peptide [142], and starch [138], have been reported, and some have undergone clinical investigation. Li et al. [138] prepared CPT-conjugated hydroxyethyl starch using glycine with two different average MWs and degrees of substitution, 130 kDa/0.4 and 200 kDa/0.5, as the spacer between the drug and polymer via ester and amide bonds. The pharmacokinetic results indicated that the biological half-life of the CPT conjugates increased from 10 minutes to 2.94 and 3.76 h, respectively.

### 4.5. Increased Specificity through Targeted Drug Delivery of PDCs

PDCs can be engineered to enhance the specificity of drug delivery to target cells or tissues through various targeting moieties. These targeting moieties can be incorporated into the PDC structure, enabling selective binding to specific receptors or markers on target cells. This targeted drug delivery approach can increase the specificity of drug delivery, minimize exposure to healthy cells, and reduce unwanted side effects. Active and passive targeting are two different strategies used in drug delivery to target specific tissues or cells in the body. Passive targeting refers to the accumulation of the drug in the target tissue or organ based on the physiological characteristics of the tissues. Cancer cells are known to often exhibit leaky vasculature and impaired lymphatic drainage, leading to a buildup of the drug in the interstitial space. This phenomenon is known as the enhanced permeability and retention (EPR) effect, and it can be exploited for passive targeting of anticancer drugs to tumor tissues. The effect of EPR is particularly advantageous for PDCs as they have a high MW and can exploit the leaky vasculature and poor lymphatic drainage of tumor tissues to accumulate in the interstitial space, resulting in increased drug accumulation and improved therapeutic efficacy [143]. Many publications have reported that PDCs can deliver a high concentration of the drug to the target site. PDCs are also effective at the cellular level. They can penetrate cells through various types of endocytosis mechanisms: phagocytosis, pinocytosis, or receptor-mediated endocytosis [144]. Active targeting, on the other hand, involves the use of ligands or targeting moieties that specifically recognize and bind to a receptor or antigen expressed on the surface of the target cells. This can improve drug accumulation in target tissues and reduce off-target effects. For example, antibodies or peptides that specifically bind to tumor cells can be conjugated to the surface of drug carriers to enhance their uptake by tumor cells [145,146]. In general, the drug-targeting efficiency of nanoparticles is achieved primarily through passive targeting, which is further increased through active targeting.

#### 4.5.1. Passive Targeting 

Publications on PDCs focus mainly on passive targeting, where the size plays a crucial role in determining the accumulation and penetration. According to reports, particle sizes within the range of 50 to 200 nm have the potential to improve the penetration of tumor tissue and increase drug accumulation in different types of cancer, including pancreatic, breast, and colon cancers. [147]. Initially, studies on anticancer drugs showed the successful development of a conjugation between neocarzinostatin and poly(styrene-co-maleic acid) (SMANCS) that could deliver to the tumor sites at a concentration higher than that of the neocarzinostatin solution. This increased accumulation of particles in the tumor sites was attributed to the EPR effect. As a result, it was approved in Japan for the treatment of hepatocellular carcinoma, and is called “Stimalmer^TM^” [148]. Laing et al. [10] conducted a study on colon cancer therapy using the conjugation of GEM and mPEG-PLA that produced small-particle-size nanoparticles (112 nm). The results of scanning electron microscopy (SEM) showed a higher accumulation of GEM in HT29 cancer cells compared to the free form of GEM. In another study, CPT was chemically linked to a biocompatible polymer, PEG [139]. The substitution of CPT was 27% *w*/*w* with a particle size of 200 nm. The biological evaluation of the PEG-CPT conjugate against HeLa cells showed improved cellular uptake and enhanced cytotoxicity compared to free CPT. Apart from particle size, surface charge plays a crucial role in influencing the EPR effect [149,150]. Within the tumor site, the surface charge affects the cellular association and the penetration effect. Cationic conjugates tend to have better interactions with negatively charged cell membranes due to electrostatic attraction, leading to increased cellular uptake [151]. Maeda et al. reported that the presence of sulfate groups and carboxylate sugars, on the luminal surface, offered a negative charge [151]. Xiao et al. reported that cationic lysines (positive charge) nanoparticles exhibited dose-dependent hemolytic activities and cytotoxicities against RAW 264.7 murine cells proportional to positive surface charge densities, while anionic aspartic acid (Asp) (negative charge) nanoparticles did not show obvious hemolytic and cytotoxic properties [152]. On the other hand, cationic conjugates can interact with negatively charged blood components, leading to aggregation or rapid clearance via the mononuclear phagocyte system (MPS) [152]. Therefore, anionic or neutral conjugates may have longer circulation times due to reduced interactions with blood components but may face some hindrance in cellular uptake due to repulsive forces between negatively charged conjugates and the cell membrane [153]. The selection of the appropriate surface charge for PDCs depends on the characteristics of the tumor microenvironment and the desired pharmacokinetics and biodistribution. Currently, numerous researchers have published work on innovative PDCs based on passive transport strategies, which are exemplified in Table 6. It should be noted that while passive targeting has shown promise in preclinical studies and some clinical trials, it may not be effective for all types of cancer or in all patients. Active targeting research continues to optimize and personalize targeted drug delivery strategies for cancer treatment.

#### 4.5.2. Active Targeting 

Active targeting involves using targeting molecules as homing devices to direct the binding of conjugates to receptor structures that are differentially expressed between normal and tumor tissues [75]. In cancer cells, several biomarkers including receptors and enzymes (e.g., folate receptor, transferrin receptor, growth factors, CD44) are overexpressed on the surface of cancer cells compared to normal cells [154,155,156,157]. Various bioactive molecules, such as antibodies, peptides, aptamers, folic acid, herceptin, and HA, have high affinity to selectively bind these receptors to cancer cells. These molecules are then conjugated to the surface of nanocarriers to act as targeting ligands. For example, Issarachot et al. [12] investigated PEG-PCL- and MTX-conjugated nanoparticles decorated with folic acid, compared with undecorated folic acid nanoparticles. The result of an in vitro uptake study showed that PDC nanoparticles decorated with 10 mol% folic acid were taken up by MCF-7 cells significantly more than undecorated folic acid nanoparticles. In another study, HA served as a natural ligand for tumor-targeted drug delivery systems, as it contains the endocytic HA receptor, CD44, which is overexpressed in many cancer cells [94]. HA-based drug nanocarriers have been used in various anticancer therapies, such as GEM-HA-DOX [158], GEM-HA-CUR [34], HA-PTX [97], HA-CPT [159], and HA-CIS [160]. Vogus et al. [158] synthesized an acid-sensitive DOX-GEM-gly-HA prodrug with different drug ratios using amide and hydrazone as chemical linkers. In vitro and in vivo studies revealed that the dual drug conjugate was more effective in inhibiting 4T1 tumor growth by 60% less than those treated with free drugs. In our previous study, GEM and CUR were successfully conjugated on HA using a hydrazone linker as a pH-sensitive linker. The results showed that HA could promote the uptake of GEM and CUR nanoparticles and provided greater cytotoxicity in HCT116 and A549 cells compared to free drugs [34]. Furthermore, one of the extensive studies in clinical research and preclinical trials focuses on the targeted drug delivery of monoclonal antibodies conjugated with drugs achieving high-potent cytotoxic effects and reducing side effects. Reynolds et al. [161] revealed that HER2-targeted liposomal doxorubicin offers a clinical advantage by enhancing the therapeutic potential of HER2-based treatments and reducing the cardiotoxicity associated with anthracyclines for HER2-overexpressing cancers. Another alternative targeting ligand, a MMP-2-sensitive copolymer, is attractive cancer biomarkers that is overexpressed in tumor tissues. Several MMP-2-sensitive copolymers have been developed for tumor targeting [162]. Yao et al. [163,164] developed MMP-2-sensitive polymeric nanoparticles of PEG-phosphoethanolamine-based copolymers (PEG-pp-PE) that showed tumor targeting and could inhibit P-gp-mediated drug efflux. The results also indicated that the P-gp inhibition capability of the PEG-pp-PE copolymers was highly associated with P-gp downregulation, an increase in plasma membrane fluidity, and inhibition of P-gp ATPase activity. Examples of PDCs based on targeted drug delivery are also shown in Table 6.

**Table 6 pharmaceutics-15-02216-t006:** PDCs based on passive and active targeted drug delivery.

Polymer Compositions	Drugs	Ligand	Linkers	Particle Size (nm)	Application	Ref.
**Passive targeting**						
mPEG-PLA	GEM	-	Amide	112.2 ± 1.86	Enhanced the efficacy and the stability of blood circulation in the animal model	[10]
PEGMA-PLA	CPT	-	Ester	37.54	Improved drug stability	[140]
PEG	CPT		Ester	171.9 ± 7.5	Improved cellular uptake Enhanced cytotoxicity	[139]
MPEG-*b*-norbornene functional PLA-*b*-P(α-BrCL)	PTX DOX	-	Ester and amide	67.8 ± 4.50	Enhanced the efficacy and synergistic effect	[165]
Galactosylated pullulan	CUR	-	Ester	355 ± 9	Enhanced cytotoxicity in hepatocellular carcinoma	[166]
Acetylated carboxymethylcellulose (Ac-CMC)	Cabazitaxel (CBZ) Docetaxel (DTX)	-	Ester	96 ± 5.3	Enhanced cytotoxicity in resistant prostate cancer	[167]
**Active targeting**						
PEG-PCL	MTX	Folic acid	Ester	200–300	Enhanced cellular uptake	[12]
Generation 5 polyamidoamine	MTX	Folic acid	Amide	-	Increased specificity Enhanced cytotoxicity in HeLa cells from cervical carcinoma	[168]
HA	DOX GEM	HA	Amide Hydrazone	20–100	Increased specificity Enhanced cytotoxicity in a 4T1 orthotopic mouse breast cancer model	[158]
HA	GEM CUR	HA	Hydrazone	221.2 ± 7.7	Increased specificity Enhanced cytotoxicity in HCT116 and A549 cells	[34]
HA	PTX	HA	Ester	-	Enhanced efficacy in mice with bladder cancer	[97]
HA	CPT	HA	Amide	-	Improved stability Enhanced cellular uptake	[159]
HA	CIS	HA	Ester	-	Enhanced cytotoxicity	[160]
PEG	DOX	TTP		-	Increased specificity Enhanced cellular uptake and efficacy	[161]
PLGA-PEG	Trastuzumab (TTP)	TTP	Amide	81.2 ± 0.9 to 102.5 ± 0.7	Reduced phagocytic uptake and immunogenicity Increased cellular uptake	[169]
PEG-PE	DOX	PEG-pp-PE (MMP-2 sensitive polymer)	Peptide	33.0 ± 1.2	Improved multidrug resistance and enhanced efficacy	[163,164]
PEG-PLA	Irinotecan (CPT-11)	PEG-pp-PLA (MMP-2 sensitive polymer)	Peptide	172 ± 30	Improved multidrug resistance and enhanced efficacy	[162]

## 5. Conclusions

The use of PDCs in pharmaceutical delivery systems enables the proper delivery of drugs and their release at the target site in prodrug form. In this review, we discuss how to modify PDCs to address pharmaceutical challenges. To achieve therapeutic efficiency, polymer structures, small molecules, targeting ligands, linkers, and drug properties should be taken into consideration, along with an understanding of the biological conditions of diseases. Various strategies can be employed to modify polymer structures, such as combining them with other polymers and/or decorating them with small molecules, to attain desired properties such as self-assembled formation, hydrophobic and hydrophilic balance, and reduced recognition by immune systems. Moreover, the choice of linkers between the drug and polymer is crucial, as they determine the timing and location of drug release under physiological conditions such as pH, temperature, enzymes, and disease-specific overexpressed receptors. These linkers, known as stimuli-responsive linkers, depend on the functional groups available in the structure of the drug and polymer, which also affect the physiochemical properties of PDCs. Furthermore, targeted drug delivery for cancer enhances both passive and active approaches, representing a significant advance in improving the specificity and efficacy of chemotherapy. All strategies should be optimized to make PDCs useful in clinical applications, offering hope for more effective and personalized cancer therapies. It is important that PDC preparation does not require complex multistep processes, is sufficiently stable during storage, and is easy to use in clinical settings. The PDC approach is fascinating and appears to have a bright future in therapeutics.

## Figures and Tables

**Figure 1 pharmaceutics-15-02216-f001:**
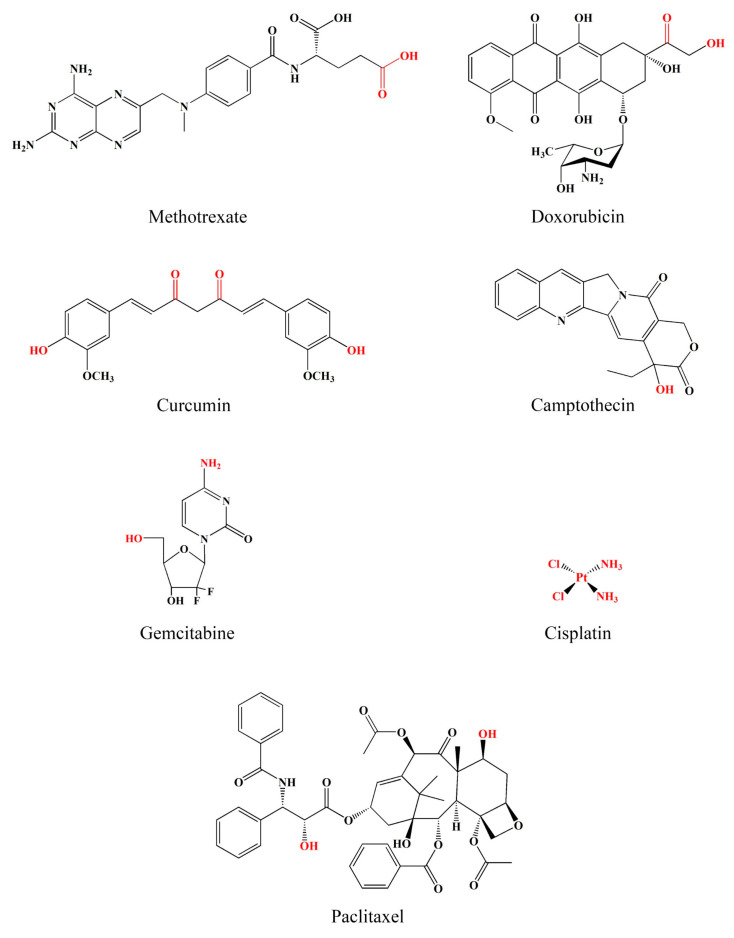
Examples of chemical structures of anticancer drugs having functional groups (red font) for possible conjugation.

**Figure 2 pharmaceutics-15-02216-f002:**
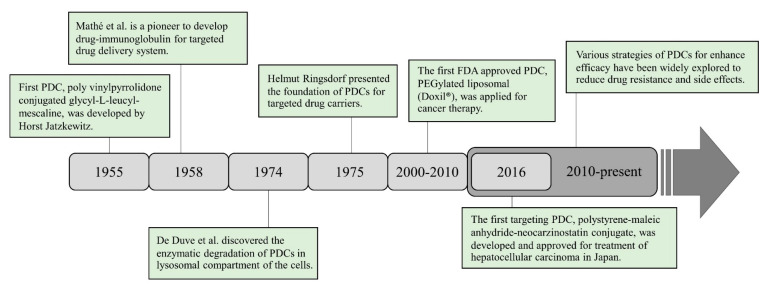
History, progress, and research stages of polymer drug conjugates.

**Figure 3 pharmaceutics-15-02216-f003:**
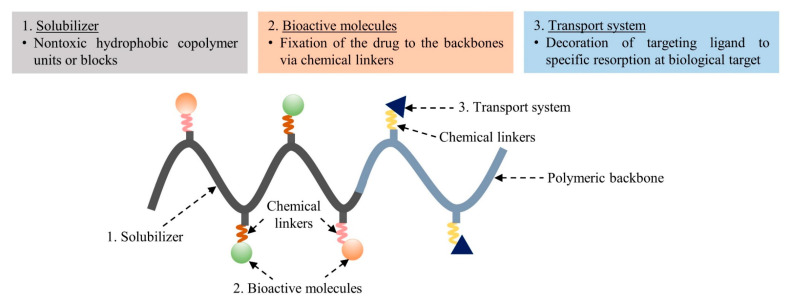
Basic concept of polymer–drug conjugates based on the Ringsdorf model.

**Figure 4 pharmaceutics-15-02216-f004:**
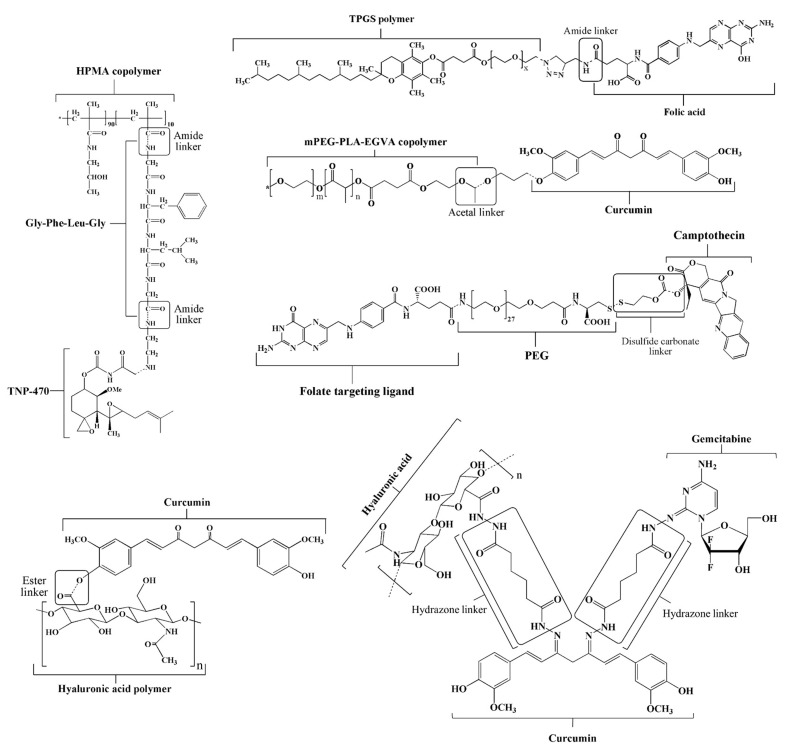
Chemical structures of polymer–drug/bioactive conjugates using various linkers. EGVA: ethylene-glycidyl methacrylate-vinyl acetate.

**Figure 5 pharmaceutics-15-02216-f005:**
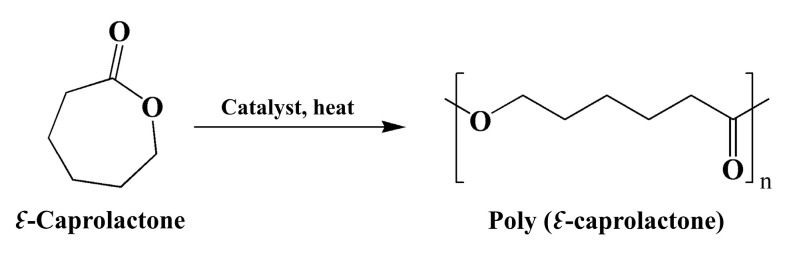
Ring-opening polymerization (ROP) of CL to obtain P(CL).

**Figure 6 pharmaceutics-15-02216-f006:**
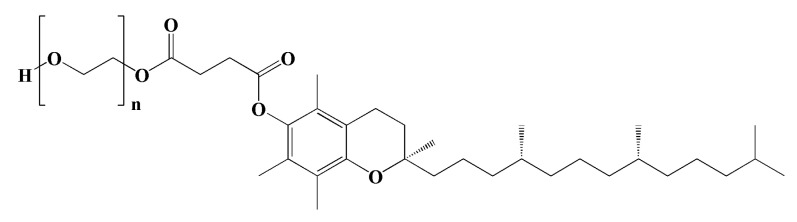
The chemical structure of TPGS.

**Figure 7 pharmaceutics-15-02216-f007:**
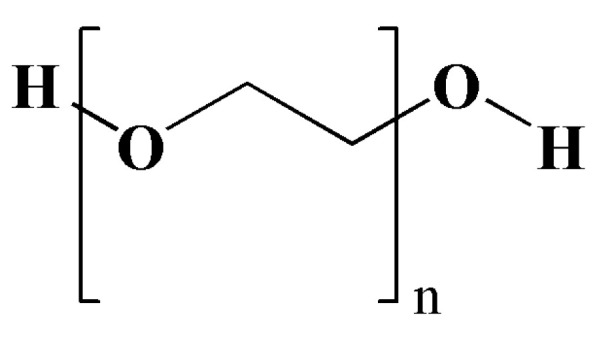
The chemical structure of PEG.

**Figure 8 pharmaceutics-15-02216-f008:**
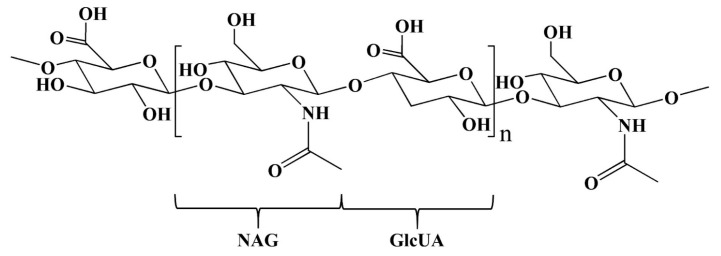
The chemical structure of HA. Abbreviations: NAG (N-acetyl-D-glucosamine) and GlcUA (D-glucuronic acid).

**Figure 9 pharmaceutics-15-02216-f009:**
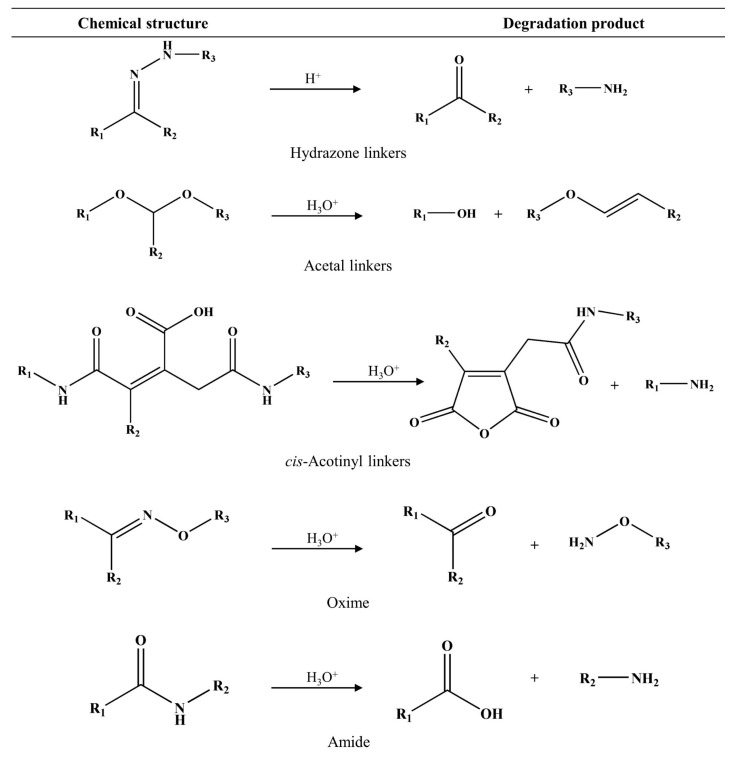
Examples of pH-sensitive chemical linkers and their degradation products.

**Figure 10 pharmaceutics-15-02216-f010:**
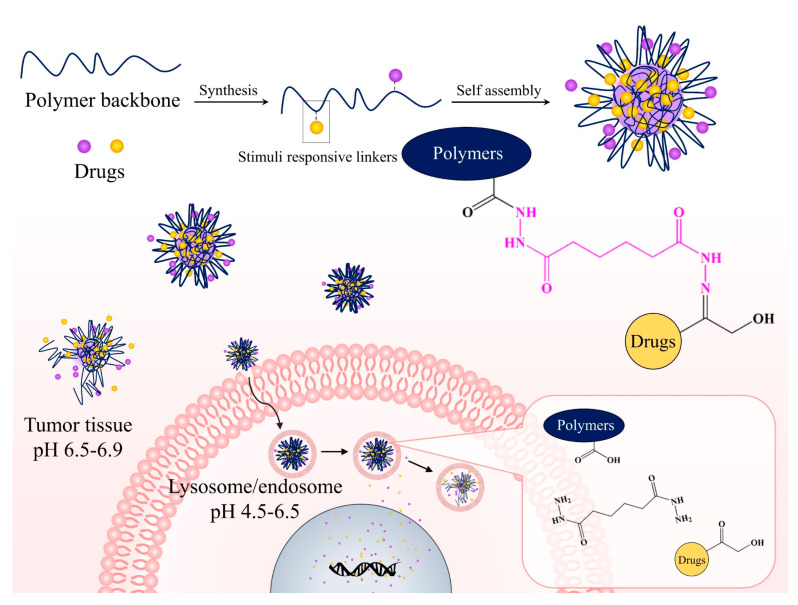
PDCs transported into cancer cells and their release using pH-sensitive linkers.

**Figure 11 pharmaceutics-15-02216-f011:**
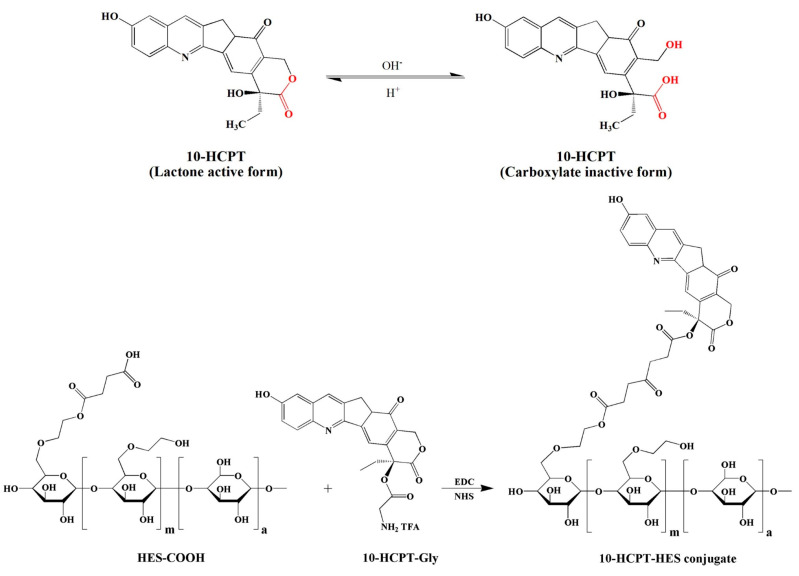
The chemical structure of CPT in three forms: lactone form, inactive carboxylate form, and conjugation process [138]. Red fonts indicate the site of lactone conversion to carboxylate form.

**Table 1 pharmaceutics-15-02216-t001:** The design of polymer–drug conjugates based on PCL backbones.

Polymer Compositions	Grafting Ligand	Drug	Disease	Application	Ref.
Folic acid-PCL-PEG	Folic acid	MTX	Breast cancer	Enhanced cytotoxicity and specificity	[12,50]
Oleic acid-PEG-*b*-PCL	Oleic acid	Curcumin (CUR)	Brain cancer	Enhanced accumulation in the brain	[52]
Folic acid-(P(CL)_2_-PEG	Folic acid	MTX	Breast cancer	Enhanced cytotoxicity and specificity	[12,50]
PCL-TPGS	-	Quercetin (QCT)	Breast cancer	Enhanced drug loading capacity Sustained drug release	[44]
Bi(mPEG-SeSe)-PCL	-	DOX	Skin cancer	Enhanced cytotoxicity and specificity	[53]

**Table 2 pharmaceutics-15-02216-t002:** The design of PDCs based on TPGS backbones.

Polymer Compositions	Grafting Ligand	Drug	Application	Ref.
TPGS		DOX	Increased drug stability Enhanced cellular uptake and efficacy Reduced side effects in vivo	[70]
TPGS		CIS	Enhanced the efficacy Presented neuroprotective effect	[60]
TPGS		GEM	Improved cytotoxicity	[71]
TPGS		DTX Cetuximab (Cmab)	Achieved synergistic effects for multidrug resistance Enhanced the efficacy	[72]
mPEG-paclitaxel/TPGS		PTX	Achieved synergistic effects for multidrug resistance Enhanced cellular uptake. Enhanced the efficacy	[73]
TPGS-*b*-PCL/Pluronic P123	Anti-GPC3 antibody	Sorafenib (Sf)	Enhanced cellular uptake and cytotoxicity in liver cancer	[68]
PLA-TPGS	Transferrin	DTX	Improved pharmacokinetic profile Enhanced cytotoxicity and efficiency in vivo	[74]
TPGS/TPGS	Folic acid	MTX	Enhanced the targeted drug delivery	[61]

**Table 3 pharmaceutics-15-02216-t003:** Summary of the main conjugates of HA drugs.

MW of HA (kDa)	Drug	Administration Route *	Disease	Tumor Model	Ref.
200	PTX	i.p. and i.v.c	Ovarian cancer, bladder cancer	OVCAR-3, SKOV-3, Phase II clinical trial	[97]
40	PTX	i.v.	Squamous cell carcinoma of the head and neck	OSC-19, NH5	[100]
5	PTX	i.v.	Brain metastasis, breast cancer	231 Br	[101]
35	DOX	s.c.	Breast cancer	MDA-MB-468LN	[99]
200	CPT	i.p.	Peritoneal cancer	HT-29, MKN-45, OE-21, DHD/K21/Trb	[102]
35	CIS	s.c.	Breast cancer	MCF-7, MDA-MB-231	[98]
10	QCT	i.v.	Hepatoma	H22	[103]
11	GEM/CUR	i.v.	Pancreatic cancer, colon cancer, lung cancer	PANC-1 Caco-2, HCT116 A549	[34]

* i.p.: intraperitoneal, i.v.c.: intravesical, i.v.: intravascular, s.c.: subcutaneous.

**Table 4 pharmaceutics-15-02216-t004:** Enhancement of drug solubility via polymer–drug conjugates.

Polymers	Drugs	%DL * (%*w*/*w*)	Solubility in Water		Application	Ref.
			Conventional	PDCs		
HA	CUR	1.3+0.31	0.27 µg/mL	7.5 mg/mL	Improved stability	[115]
PEG	PTX	60.3	<2 μg/mL	3665 μg/mL	Human cervical carcinoma	[11]

* %DL: drug loading capacity.

**Table 5 pharmaceutics-15-02216-t005:** The development of PDCs to enhance controlled and sustained release through enzyme-sensitive linkers.

Polymer Compositions	Drugs	Specific Enzymes	Stage	Application	Ref.
HPMA-Gly-Phe-Leu-Gly	-	Cathepsin B	In vitro	Increased stability in plasma	[130]
Brentuximab vedotin-Val-Cit-PABC	Monomethyl auristatin E	Cathepsin B	FDA approval	Used for Hodkin lymphoma	[131]
PEG-Gly-Cys-Gly-Ala-Ala-Asn-Leu-Glu	CIS	Legumain	In vitro	Increased drug stability in plasma; enhanced gastric cancer therapy	[125]
PEG-Gly-Phe-Leu-Gly	GEM	Cathepsin B	In vitro and in vivo	Increased drug stability in plasma; increased antitumor activity in breast cancer, but reduced side effects to normal tissues	[126]
NTD-Gly-Phe-Leu-Gly	DOX	Cathepsin B	In vitro	Increased stability in plasma; enhanced drug accumulation in liver cancer cells	[132]
Dextran-Pro-Val-Gly-Leu-Ile-Gly	MTX	MMP-2/MMP-9	In vitro	Increased stability in plasma; enhanced drug accumulation in fibrosarcoma cell line and liver cancer cells	[129]
HPMA-morpholinocarbonyl-Ser-Ser-Lys-Tyr-Gln-Leu	12-aminododecanoyl thapsigargin	Cathepsin B	In vitro and in vivo	Enhanced drug accumulation in prostate cancer cells	[133]

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
