# Peer review of "Innovative Design of Targeted Nanoparticles: Polymer–Drug Conjugates for Enhanced Cancer Therapy"

_pharmaceutics, 2023, doi:10.3390/pharmaceutics15092216_

Round 1
Reviewer 1 Report
Point 1: Sections 1 and 2 (that is, the first 3 pages) contain common explanations. Section 3 describes the general structure of the PDC, and Section 3.1 describes the specific structure.
I would like you to combine Sections 1 and 2 to create an introduction that explains the advantages, necessity, and history of a general PDC. I also suggest including a timeline figure that displays important events in the PDC history up to 2023. Before discussing specific polymers, it would be helpful to classify and mention the polymers covered in the review.
Point 2. At the beginning of the text, the basic components and structures related to PDC composition are explained first (Fig.2), and the presence of functional groups of drugs to form PDC is also displayed next (Fig1). In particular, paclitaxel (PAC), which is mentioned a lot in the text, is missing from Fig.1, so it is recommended that PAC's structure and functional group be included in the figure.
Point 3: The authors give Figure 3 with the chemical structures of polymer-drug/bioactive conjugates using various linkers. Since there is a poor description of the figure and discussion in section 3.1, and an explanation appears in section 3.2, it is recommended to reorganize the text and figure to establish a connection or indicate a related figure in 3.2.
Point 4. Among the PDCs covered in the review, it would be nice to have a table that classifies and organizes FDA-approved drugs and clinical progress. Although there are limitations of PEG, the discussion needs to be presented on why polymers other than PEG have limitations in FDA approval.
Point 5. In Section 3.3.2, it has been noted that MMPs are shown in Table 5 but are not described in the text. Therefore, it is recommended to mention and explain the MMPs in the text as well.
Point 6. I have reviewed the sentences in lines 515-520 of Section 3.5.1. The first focus was on targeted delivery by linking antibodies, but does that mean that passive delivery by nano-sizing is better than targeted delivery? The description is ambiguous and requires clarification.
Point 7. On line 528, it is Table 6, not Table 5.
Point 8. It would be better if the examples displayed in Table 6 were thoroughly explained and discussed in the text.
Point 8. In the case of PAC, DOX, CPT, CIS, and curcumin, which have been studied a lot, they would have been produced using various types of polymer conjugation. I suggest adding a table describing the anticancer efficiency of a drug depending on polymer conjugation.
Through this, it will be possible to compare which PDC depending on the drug, has a better effect on which cancer.
Point 9. Author contributions and funding have not been made.
Point 10. The percentage of papers from the last five years in the reference list is not high. It is necessary to understand how PDCs have been developed and utilized through recently published paper reviews.
Reviewer 2 Report
The authors reviewed the recent advances in polymer-drug conjugates for cancer therapy. Moreover, it reports the polymers, the conjugating strategies, and targeting ligands for PDC preparation. This manuscript is orderly arranged. I have some minor comments before accepting it for publication.
1. The resolution of Fig. 2 and Fig.4 is insufficient to publish in the journal. Please change it to a higher resolution
2. In part 3.1, only four polymers were discussed in preparing PDC. Other materials are needed, for example, proteins.
The language in this manuscript needs polishing.
Reviewer 3 Report
The manuscript titled as Innovative design of targeted nanoparticles: Polymer-drug con- 2 jugates for enhanced cancer therapy with Manuscript Number Pharmaceutics-2484317 submitted to Pharmaceutics for possible publication is not suitable for the publication as such due to following points. Manuscript should be revised. Major Revisions.
1. The abbreviation used should be explained when firstly used.
2. There are lot of review articles on the topic and increasing in the number with passage of time…Highlight the difference/novelty of this review from already published ones.
3. References should be updated as only 2 references from 2023 and 4 from 2022. Even the WHO website was accessed in February 2022…….Updated data should be added
Authors should try to remove the minor mistakes in Sentence building.
Reviewer 4 Report
The authors reviewed the recent advances in polymer-drug conjugates for cancer therapy on title “Innovative design of targeted nanoparticles: Polymer-drug conjugates for enhanced cancer therapy”. Moreover, this manuscript reviewed and reports the polymers, conjugating strategies, and targeting ligands for PDC preparation especially designed in cancer therapy. This manuscript is orderly arranged and well written and may be acceptable with minor revision.
· Complex sentences must be simplified (introduction part row 34-37, 40-46)
· Repeated “the” in single sentences shall be removed (Introduction part, row 98-102, 137- 139) .
· Line 129 mentioned “PEG-lasparaginase” ….?? must be in proper format
· Fig 3 EGVA abbreviation must be included
· Tabular arrangement (Table no. 1 and Table no. 2) is confusing. Why author mentioned the Naproxen research studies in Table 1, similarly ibuprofen, clodronate etc as this polymer-drug conjugate for the cancer not the rheumatoid arthritis. See the table 2 as why author summarized the studies of itraconazole as a fungal indication. In my opinion this manuscript, cover the polymer-drug conjugates strategy for cancer therapy not for other therapeutic indication.
· Complex sentences must be simplified (row 190-193, 253-256).
· Row (259-262) “D-glucuronic acid (GlcUA) linked through β-1,4-linkage and N-acetyl-D-
· glucosamine (NAG) via β-1,3-linkage” so it must be mentioned respectively.
· Tabular arrangement (Table no.3 and 4) should not be confusing.
· Line 313-315 rewrite this sentence with proper grammatical arrangement.
· Too long sentences must be simplified and corrected properly (e.g. line 361-364).
· Sub-heading 3.3, I will recommend to add few case studies to support this point
· Tabular arrangement (Table no.5 and 6) must be clear and mentioned text formatting should be in sentence case.
· Heading 3.1 mentioned “PDC Architectures” main heading shall be named to cover all sub-heading.
· Author contribution and funding information should be there in manuscript.
· Active and passive targeting approaches must be concluded in conclusion part shortly.
· The resolution of Fig. 2 and Fig.4 is not better and recommended to revise with high resolution.
· In part 3.1, only four polymers were discussed in preparing PDC. Other materials are needed, for example, proteins.
· Figure 1. Why ibuprofen and Sulfasalzine, dapsone shows as this manuscript is for the cancer therapy only.
Some grammar and spelling check is required.
Refere the following paper also in reference;
https://www.sciencedirect.com/science/article/pii/S1359644620302567?via%3Dihub
Ekladious, I., Colson, Y.L. & Grinstaff, M.W. Polymer–drug conjugate therapeutics: advances, insights and prospects. Nat Rev Drug Discov 18, 273–294 (2019). https://doi.org/10.1038/s41573-018-0005-0
Pradip Thakor, Valamla Bhavana, Reena Sharma, Saurabh Srivastava, Shashi Bala Singh, Neelesh Kumar Mehra, Polymer–drug conjugates: recent advances and future perspectives, Drug Discovery Today, Volume 25, Issue 9, 2020, Pages 1718-1726, https://doi.org/10.1016/j.drudis.2020.06.028.
NA
Round 2
Reviewer 1 Report
Minor revision
The paper has been revised in accordance with the recommendations of the reviewer, resulting in a significant improvement from its previous version.
However, the following points must be corrected before publication.
Point 1. Section 5. Acknowledgment, 6. Author statement, and 7. References should be sub-headlined and subsequent appropriate content.
Point 2. Check References number. The numbers have been pushed one by one. So numbers in parentheses in the body text and reference numbers do not match.
Reviewer 3 Report
Dear Authors,
Thanks for revising your manuscript and improving it. However, some minor mistakes are still there like;
1. Please confirm that abbreviation CUR is completely explained when firstly used in Introduction section.
2. Spacing of line 450-453 should be addressed. 461-462 font style issues.
3. There should be no space between sign of degree centigrade and the value.
4. Heading of references is considered as 1st reference. References should be set accordingly..In this way whole numbering is incorrect.
5. Figure 5 is not mentioned in the text.
Minor issues in sentence building.
